# Risk Score Generated from CT-Based Radiomics Signatures for Overall Survival Prediction in Non-Small Cell Lung Cancer

**DOI:** 10.3390/cancers13143616

**Published:** 2021-07-19

**Authors:** Viet-Huan Le, Quang-Hien Kha, Truong Nguyen Khanh Hung, Nguyen Quoc Khanh Le

**Affiliations:** 1International Master/Ph.D. Program in Medicine, College of Medicine, Taipei Medical University, Taipei 110, Taiwan; d142109009@tmu.edu.tw (V.-H.L.); m142109004@tmu.edu.tw (Q.-H.K.); d142108005@tmu.edu.tw (T.N.K.H.); 2Department of Thoracic Surgery, Khanh Hoa General Hospital, Nha Trang City 65000, Vietnam; 3Department of Orthopedic and Trauma, Cho Ray Hospital, Ho Chi Minh City 70000, Vietnam; 4Professional Master Program in Artificial Intelligence in Medicine, College of Medicine, Taipei Medical University, Taipei 106, Taiwan; 5Research Center for Artificial Intelligence in Medicine, Taipei Medical University, Taipei 106, Taiwan; 6Translational Imaging Research Center, Taipei Medical University Hospital, Taipei 110, Taiwan

**Keywords:** non-small cell lung cancer, radiomics radiology, overall survival, prognostic biomarkers, multivariate analysis

## Abstract

**Simple Summary:**

Despite recent advancements in lung cancer treatment, individuals with lung cancer have a dismal 5-year survival rate of only 15%. In patients with non-small cell lung cancer (NSCLC), medical images have lately been employed as a valuable marker for predicting overall survival. The primary goal of this study was to develop a risk score based on computed tomography (CT) based radiomics feature signatures that may be used to predict survival in NSCLC patients. After analyzing 577 NSCLC patients from two data sets, we discovered that the risk score model’s prediction ability as a prognostic indicator was superior to other clinical indicators (age, stage, and gender), and the possibility of patient risk stratification with survival was evaluated using a risk score representation of 10 radiomics signatures. According to this study, the risk score generated using CT-based radiomics signatures promises to predict overall survival in NSCLC patients.

**Abstract:**

This study aimed to create a risk score generated from CT-based radiomics signatures that could be used to predict overall survival in patients with non-small cell lung cancer (NSCLC). We retrospectively enrolled three sets of NSCLC patients (including 336, 84, and 157 patients for training, testing, and validation set, respectively). A total of 851 radiomics features for each patient from CT images were extracted for further analyses. The most important features (strongly linked with overall survival) were chosen by pairwise correlation analysis, Least Absolute Shrinkage and Selection Operator (LASSO) regression model, and univariate Cox proportional hazard regression. Multivariate Cox proportional hazard model survival analysis was used to create risk scores for each patient, and Kaplan–Meier was used to separate patients into two groups: high-risk and low-risk, respectively. ROC curve assessed the prediction ability of the risk score model for overall survival compared to clinical parameters. The risk score, which developed from ten radiomics signatures model, was found to be independent of age, gender, and stage for predicting overall survival in NSCLC patients (HR, 2.99; 95% CI, 2.27–3.93; *p* < 0.001) and overall survival prediction ability was 0.696 (95% CI, 0.635–0.758), 0.705 (95% CI, 0.649–0.762), 0.657 (95% CI, 0.589–0.726) (AUC) for 1, 3, and 5 years, respectively, in the training set. The risk score is more likely to have a better accuracy in predicting survival at 1, 3, and 5 years than clinical parameters, such as age 0.57 (95% CI, 0.499–0.64), 0.552 (95% CI, 0.489–0.616), 0.621 (95% CI, 0.544–0.689) (AUC); gender 0.554, 0.546, 0.566 (AUC); stage 0.527, 0.501, 0.459 (AUC), respectively, in 1, 3 and 5 years in the training set. In the training set, the Kaplan–Meier curve revealed that NSCLC patients in the high-risk group had a lower overall survival time than the low-risk group (*p* < 0.001). We also had similar results that were statistically significant in the testing and validation set. In conclusion, risk scores developed from ten radiomics signatures models have great potential to predict overall survival in NSCLC patients compared to the clinical parameters. This model was able to stratify NSCLC patients into high-risk and low-risk groups regarding the overall survival prediction.

## 1. Introduction

Lung cancer is still the most common cause of death globally, as well as the second most common cancer, according to Sung et al. [1], with mortality and occurrence rates of 11.4% and 18%, respectively. Lung cancer incidence and mortality have increased significantly globally in the last few years, both in absolute and relative terms. Two hundred and twenty-eight thousand, eight hundred and twenty new lung cancer cases and more than one hundred thousand deaths from lung cancer were expected to occur in 2020 [1]. Although recent advances in lung cancer treatment, lung cancer patients have a disappointing 5-year survival rate of just 15% [1].

Non-small cell lung cancer (NSCLC) is the most common form of lung cancer, accounting for 85% of all cases [2]. The Tumor–Nodal–Metastasis (TNM) grading system is still the primary method for predicting NSCLC patients’ survival time; in addition, age and gender are also prognostic factors. While TNM is a commonly used method, it has some drawbacks when the staging is the same, but the survival times are different [3]. Therefore, it is necessary to develop additional tools to help predict and improve overall survival in patients with NSCLC.

Medical images have recently been used as a helpful marker. It is a commonly used diagnostic tool that is convenient, highly reliable, and non-invasive. Radiomics is a non-invasive diagnostic tool that uses quantitative features extracted from medical imaging data, such as computed tomography (CT), to make diagnoses, which is a strong imaging marker for prognosis in NSCLC patients [4]. Association between radiomics and overall survival in NSCLC patients has been studied in recent studies [5,6,7,8,9,10,11,12]. Some authors developed CT-based radiomics signatures for survival prediction in NSLCL patients [6,7,8,9,10,13,14,15,16]. Among these studies, Aerts et al. [13] found four feature signatures representing four main groups of features: original, tumor intensity, first order, and wavelet, and used these four features in turn to evaluate the effectiveness of predicting survival in NSCLC patients in the training set. Most of the authors work towards finding sets of radiomics features closely related to survival prediction, called sets of radiomics signatures, and used them for survival prediction for cancer patients [5,6,9,10,12,14,15,16]. Similarly, some authors used the expression of a group of signature genes to predict survival in cancer diseases; some constructed risk scores to represent sets of signature genes to optimize survival prediction in cancer patients [17,18,19,20,21]. It is also necessary to use a representative scale score similar applied to radiomics field.

We want to study how effective one scale score that is representative of the feature signatures found is for predicting survival in NSCLC patients and compare the effectiveness of this score with clinical parameters. In our study, from the risk score created based on the radiomics features signatures, we hypothesize that the risk score developed from the radiomics signatures model can predict overall survival in NSCLC patients more accurately and precisely than the clinical parameters. This study aims to develop and assess the overall survival prediction ability of the risk score to contribute to improving predictive survival of NSCLC patients compared to other clinical parameters, such as stage, gender, age.

## 2. Results

### 2.1. Patient’s Clinical Characteristics

We analyzed a total of 577 NSCLC patients from two data sets. The first data set is Lung 1 (NSCLC-Radiomics); it consists of 420 NSCLC patients. The second dataset is Lung 2 (NSCLC Radiogenomics), which consists of 157 Non-Small Cell Lung Cancer (NSCLC) patients. Evaluation of demographic characteristics between two datasets showed no difference in terms of age (*p* = 0.302), gender male (*p* = 0.967), but there was a difference in the type of NSCLC, survival time, stage (*p* < 0.001) (Table 1).

We randomly divided Lung 1 data set into two parts: 336 patients in the training set group, 84 patients in the testing set group. The mean age was 67.67 ± 10.04 (standard deviation) in the training set and 69.36 ± 10.19 (standard deviation) in the testing set. The second data set included 157 patients for the external validation set group (mean age 68.97 ± 9.52 (standard deviation)). There was no difference between characteristics of age (*p* = 0.183), gender male (*p* = 0.654), type of NSCLC (*p* = 0.545), stage (*p* = 0.126), and survival time (*p* = 0.073) between the training set and testing set (Table 1). The training set had a median of overall survival of 579.00 ((286.25, 1510.25) (IQR)), in testing set it was 456.50 ((229.00, 1038.50) (IQR)), and validation set was 1315.00 ((630.00, 1921.00) (IQR)). The cell types that make up the majority of the training set were Squamous cell (115 (38.3%)) and Large cell (95 (31.7%)) and Squamous cell (36 (46.2%)) and Large cell (19 (24.4%)) in the testing set. Adenocarcinoma was the most frequent cell type in the validation set (126 (80.3%)). The majority of patients were in stage IIIb in the training and testing set, and stage I accounted for the majority in the validation set.

### 2.2. Feature Selection

In the training set, 136 radiomics features were retained after excluding the redundant features by Pearson’s correlation pairwise selection (Figure 1A). We built the LASSO model to figure out the best features for predicting overall survival. In the LASSO regression model, there were 13 features with non-zero coefficients and then 13 best features to continue the process of finding radiomics features’ signatures (Figure 1B).

### 2.3. Construction of Ten Radiomics Signatures Prognostic Model

Univariate Cox regression analysis was performed to establish the best model for survival prediction in the training set. Ten features were significantly associated with predicting overall survival in the training set (Table 2): original.shape.Elongation (*p* = 0.04), original.gldm.DependenceVariance (*p* = 0.02), wavelet.LHL.firstorder.Skewness (*p* < 0.01), wavelet.LHH.gldm.LargeDependenceHighGrayLevelEmphasis (*p* < 0.001), wavelet.LHH.firstorder.Mean (*p* = 0.02), wavelet.LLH.glcm.ClusterShade (*p* = 0.04), wavelet.LLH.firstorder.Maximum (*p* < 0.01), wavelet.HLH.firstorder.Energy (*p* < 0.001), wavelet.HLH.firstorder.Skewness (*p* < 0.01), wavelet.HLL.glszm.GrayLevelNonUniformityNormalized (*p* < 0.01).

### 2.4. Risk Score Establishment for Overall Survival Prediction in the Training Set

To comprehensively explore the relationship between these ten identified radiomics features and the survival prognosis in NSCLC patients, we developed a risk score generated from ten radiomics signatures based on their Cox coefficients (Figure 2) (Table 3). We measured the risk score for each patient and rated them based on it.

Risk score = (−9.936 × 10^−1^ × original.shape.Elongation) + (5.230 × 10^−3^ × original.gldm.DependenceVariance) + (−1.693 × 10^−1^ × wavelet.LHL.firstorder.Skewness) + (4.764 × 10^−6^ × wavelet.LHH.gldm.LargeDependenceHighGrayLevelEmphasis) + (−1.489 × 10^−1^ × wavelet.LHH.firstorder.Mean) + (−6.460 × 10^−6^ × wavelet.LLH.glcm.ClusterShade) + (1.074 × 10^−4^ × wavelet.LLH.firstorder.Maximum) + (8.738 × 10^−9^ × wavelet.HLH.firstorder.Energy) + (3.567 × 10^−1^ × wavelet.HLH.firstorder.Skewness) + (2.988 × wavelet.HLL.glszm.GrayLevelNonUniformityNormalized).

The results of the investigation into the correlation between the proposed radiomic risk score showed the ten radiomics signature features with tumor volume were low (r < 0.75) (Figure 3). The means of the ten radiomics signatures and the risk score were consistent and signified independent predictability for overall survival prediction in this study.

Patients were classified into high- and low-risk groups based on a cutoff of the median of the risk score, which has different expression patterns to the radiomics features’ signatures. We investigated the difference between the two groups and overall survival in NSCLC patients after dividing them into high- and low-risk groups. The high-risk score patients group had a lower survival time than the low-risk patient’s group, according to the Kaplan–Meier curve (log-rank test, *p* < 0.001) (Figure 4A).

The prediction ability of the risk score was 0.696 (95% CI, 0.635–0.758), 0.705 (95% CI, 0.649–0.762), 0.657 (95% CI, 0.589–0.726) (AUC) for 1, 3, and 5 years in the training set group (Figure 4B). After constructing a ten radiomics model, a multivariate Cox proportional hazard model was also conducted to confirm the model’s capacity to predict prognosis independently. The findings revealed that the risk score model could predict overall survival in NSCLC patients regardless of stage, gender, and stage (HR, 2.99; 95% CI, 2.27–3.93; *p* < 0.001) in the training set. The hazard ratio value of the risk score was more than one and greater than the hazard ratio values for age, gender, and stage, implying that NSCLC patients who have a high-risk score would have a poor prognosis (Figure 5A).

When comparing the prediction ability of the risk score with clinical parameters, such as age, gender, and stage, in the training set, the risk score was more likely to have better accuracy in predicting survival at 1, 3, and 5 years than clinical factors. In training set, risk score AUC was 0.696 (95% CI, 0.635–0.758), 0.705 (95% CI, 0.649–0.762), 0.657 (95% CI, 0.589–0.726) for 1, 3, and 5 years compared with clinical factors, such as age 0.57 (95% CI, 0.499–0.64), 0.552 (95% CI, 0.489–0.616), 0.621 (95% CI, 0.544–0.689); gender 0.554, 0.546, 0.566; stage 0.527, 0.501, 0.459, respectively, in 1, 3 and 5 years (Figure 5B).

### 2.5. Testing and Validation of the Risk Score Model for Overall Survival Prediction

With the best signature model found in the training set, we applied it in the testing and validation set to evaluate its performance. Each patient in the testing and validation set also had one risk score generated from ten radiomics signatures, and then patients were divided into two groups: high-risk and low-risk. In the testing set, NSCLC patients with a high-risk score had a lower survival time than the low-risk patient’s group, according to the Kaplan–Meier analysis (*p* = 0.003) (Figure 6A), and the prediction ability of the risk score was 0.758 (95% CI, 0.657–0.859), 0.764 (95% CI, 0.641–0.888), 0.719 (95% CI, 0.521–0.916) (AUC) for 1, 3, and 5 years (Figure 6B). In the validation set, the Kaplan–Meier curve separated patients into high-risk, and low-risk groups (*p* = 0.0033) (Figure 7A), and the prediction ability of the risk score was 0.676 (95% CI, 0.534–0.821), 0.629 (95% CI, 0.519–0.741), 0.709 (95% CI, 0.607–0.811) (AUC) for 1, 3, and 5 years (Figure 7B).

## 3. Discussion

Many studies have been performed by many authors developing CT-based radiomics signatures for survival prediction [6,7,8,9,10,13,14,15,16]. In our study, we aimed to see if a single scaled score representing the feature signatures discovered can be used to predict survival in NSCLC patients. The main goal was to create a risk score based on radiomics features’ signatures, which had an essential role in predicting survival in patients with non-small cell lung cancer.

Aerts et al. [13] used the same dataset (Lung 1 NSCLC-Radiomics) as their training set. They divided four main feature groups and conducted robust feature selection analysis on each group, four feature signatures representing four main groups: original, tumor intensity, first order, and wavelet, and used these four features in turn to evaluate the effectiveness of NSCLC patients in the training set. The advantage of this approach is that it can find four candidates representing four groups of radiomics; however, robust features exist randomly in four feature groups that will lead to missing out robust features if we choose only one representative feature for one group. To avoid this situation, we chose the layer-by-layer data preprocessing approach like most other authors; all the radiomics were included in the selection model initially, ensuring that robust radiomics features were chosen regardless of which group they belonged to. Because we had high-dimensional data and an enormous amount of radiomics features used as predictors in the model, and the Cox proportional hazard model often produces overfitting [22]. To avoid overfitting, we applied pairwise Pearson’s correlation feature selection method [15]. Pearson’s feature selection approach is a method that can reduce overfitting by eliminating redundant feature interactions while remaining computationally efficient [15,23]. Then the LASSO regression model helped to preserve radiomics features that were most strongly linked to overall survival [9,24]; this method was also used by another author for features selection [8,25]. One hundred and thirty-six candidate features after Pearson’s feature selection were reduced to thirteen potential radiomics features using the LASSO. All these thirteen potential radiomics features were taken into account in the univariate Cox regression model for selecting ten radiomics feature signatures based on *p*-value (<0.05). Our results showed that most of the signature features we found that strongly related to survival prediction were wavelet features. Yang, L et al. [9] analyzed the same data set (Lung 1 NSCLC-Radiomics) and showed similar results, and this finding is in line with what has been found in earlier research [26,27].

We used the risk score as an independent predicting indicator of overall survival in NSCLC patients. The predictive performance of clinical parameters in some studies shows results not relatively high compared to other parameters [12,28]. In our study, the ROC curve showed that the risk score model’s predictive performance as a prognostic indicator was superior to other clinical parameters (age, stage, and gender). In this study, the forest plot (Figure 5A) shows that the risk score could predict overall survival in NSCLC patients compared with stage, gender, and stage. Our results are similar to Hailin Li et al. [14], which showed that the signatures’ score of the radiomic features demonstrated better prognostic efficacy to the clinical characteristics (gender, age, and smoking status), and the combined model (radiomic features combined with clinical parameter).

The Kaplan–Meier curve was used to evaluate the effectiveness of patient stratification into two high-risk and low-risk groups in comparing the survival time of NSCLC patients [6,7,8,9,10,13,14,15,16]. We created a risk score representation of ten radiomics signatures for evaluating the possibility of patient risk stratification with survival and outcome results were statistically significant. Hailin Li et al. [14] also gave similar results with the radiomics signatures score representing three feature signatures to help stratify patients into two groups: high signature value and low signature value.

After having the best model, we conducted testing and validation sets. The Kaplan–Meier survival curve demonstrated a difference in overall survival between high-risk and low-risk patients (*p* < 0.05). Moreover, the risk score’s ability to predict survival at 1, 3, and 5-year survival showed similar results. When the risk score’s ability to predict survival at 1, 3, and 5-year was compared between the three data sets, training, testing, and validation, show the risk score had AUC 0.696 (95% CI, 0.635–0.758), 0.705 (95% CI, 0.649–0.762), 0.657 (95% CI, 0.589–0.726) in the training set, 0.758 (95% CI, 0.657–0.859), 0.764 (95% CI, 0.641–0.888), 0.719 (95% CI, 0.521–0.916) in the testing set and in the validation set, 0.676 (95% CI, 0.534–0.821), 0.629 (95% CI, 0.519–0.741), 0.709 (95% CI, 0.607–0.811).

Despite these promising results, there are still some limitations that can open further research. First, we only compared the effect of the risk score model and clinical parameters on survival prediction ability. Although the risk score survival prediction ability potential in NSCLC patients was higher than the clinical parameter, it was not high. We are considering integrating the radiomics model and clinical parameters in future studies. Second, according to some research, radiomics extracted using Deep Learning would be more efficient than radiomics extracted using conventional methods [11,29,30]. Based on these findings, in our following study, the prediction of overall survival could use the radiomics extract by using Deep Learning to assess the capacity to assess survival through risk score building. Third, in this study, the data came from two different institutions (Lung 1-Radiomics and Lung 2 Radiogenomics) so that there was a difference in CT image characteristics, initial image processing, segmentation, etc. The data set used for training and testing included radiomics features extracted from CT images using the manual segmentation method. In contrast, the data set used for validation was automatically segmented from CT images and checked by the experts with final approval by the thoracic radiologist. We did not perform CT images segmentation processing ourselves, which hampers the radiomics model analysis in our study. However, the data sets we used in this study are published online and have been processed of the CT data and segmentation by the author; based on these data, we used 3D Slicer software and the Pyradiomics module in Python for features extraction partially guarantee the results of the study. However, these methods do not fully guarantee the harmonization of the radiomics features extracted from these two data sets. To avoid this problem and ensure consistency, consistent data preprocessing methods should be implemented across the study, and the processing of the CT data and segmentation data should be considered before extracting radiomics features (e.g., interpolation, re-segmentation, normalization, binning, etc.), which could give more reliable and reproducible results in the following studies.

Recent studies applied low-dose CT scan images in cancer screening based on radiomics signature features [31,32]. For further research, we will consider using risk scores extracted from radiomics signatures from low-dose CT images to assess lung cancer patients’ treatment efficacy and progression. The follow-up process after treatment of lung cancer patients requires performing repeat imaging modalities (CT scan, PET/CT, etc.), so a low-dose CT scan brings many benefits to patients.

## 4. Materials and Methods

Figure 8 illustrates the workflow of our study. All the steps are described at the following sections.

### 4.1. Patients Cohort

We retrospectively collected data from two published data sets from The Cancer Imaging Archive (TCIA). The first data set was Lung 1 (NSCLC-Radiomics), which consists of 422 NSCLC patients, which was published on The Cancer Imaging Archive (TCIA) Public Access on 2 July 2014 (https://wiki.cancerimagingarchive.net/display/Public/NSCLC-Radiomics). This data set contained a manual delineation of the 3D volume of the primary gross tumor volume and selected anatomical structures by a radiation oncologist. CT data: a spiral CT, slice thickness of 3 mm and an X-ray tube current of 40–553 mA (mean 80 mA) at median peak tube voltage and 8 currents at 120 kVp (range of 120–140 kVp), with or without intravenous contrast was performed covering the complete thoracic region [13]. Due to missing radiomics features extraction information, we selected 420 cases included in this study. This data was divided randomly into 80% (336 patients) for training and 20% (84 patients) for testing, respectively (Table 1).

The second dataset was Lung 2 (NSCLC Radiogenomics), which consists of 211 Non-Small Cell Lung Cancer (NSCLC) patients in Public Access on 22 December 2015 (https://wiki.cancerimagingarchive.net/display/Public/NSCLC+Radiogenomics); CT data: slice thickness of 0.625–3 mm and an X-ray tube current of 124–699 mA (mean 220 mA) at 80–140 kVp (mean 120 kVp). Initial segmentations were obtained using an unpublished automatic segmentation algorithm from an axial CT image series. These segmentations were reviewed and edited using ePAD by a thoracic radiologist with more than five years of experience. An additional thoracic radiologist reviewed the final segmentations; disagreements in tumor boundaries were discussed and edited appropriately, with final approval by the thoracic radiologist. All segmentations are stored as DICOM Segmentation Objects [33]. We chose 157 patients to include in this study due to missing radiomics features extraction information. We used this data set for validation.

Both data sets have clinical information, including survival outcomes (overall survival), which are the times in days from CT scan to the day of death, or the date of last known alive (i.e., censored). Baseline patient characteristics are provided in Table 1.

### 4.2. Radiomics Features Extraction

We took advantage of 3D Slicer software (version 4.10.2; released on 10 October 2012; last updated on 17 May 2019), which was used for medical image extraction and visualization, to extract the features from each single CT image assay, via seven extensions (e.g., DCMQI, PETDICOMExtension, Quantitative Reporting, SlicerDevelopmentToolbox, SlicerRadiomics, SlicerRT), and the Pyradiomics [34] module in Python (version 3.8). Most radiomics features complied with the IBSI standards by using the Pyradiomics module in Python for features extraction. Initially, the CT records were imported into the software in an orderly manner for extraction. The eight hundred and fifty-one features could be categorized into four main groups: tumor intensity, shape, texture, and wavelet filters. The features were classified into nine subcategories, e.g., original, wavelet HHH, wavelet HHL, wavelet HLH, wavelet HLL, wavelet LHH, wavelet LHL, wavelet LLH, and wavelet LLL. Each category comprised 6 subcategories, namely first-order, Gray Level Co-occurrence Matrix (GLCM), Gray Level Size Zone (GLSZM), Gray Level Run Length Matrix (GLRLM), Neighboring Gray Tone Difference Matrix (NGTDM), Gray Level Dependence Matrix (GLDM), except for the original radiomics category with one sub-categorize (Shape) more than the others. The information about the radiomics classes was concretely described by Zwanenburg et al. [35]. We have also provided the radiomics features for training, testing, and validation sets in Appendix A, respectively.

### 4.3. Feature Selection and Construction of the Best Model for Survival Prediction

#### 4.3.1. Removing Highly Correlated Features in the Training Set

In order to prevent over-fitting or bias when performing analysis, we applied pairwise correlations between the radiomics features variables in the training set to exclude the redundant variables because many radiomics variables are often correlated with each other. We evaluate every pair of variables calculated by Pearson’s correlation analysis to identify the redundant features for all radiomics features in the training set [14,36], with a threshold of 0.75. If the coefficient correlation between two variables was more than 0.75, it indicated a strong correlation. Only one variable stayed in the model, so consideration was given to which had a higher correlation coefficient with the target variable (survival outcome variable), the variable with the higher correlation coefficient stayed and the others were excluded. Then, using the glmnet package in the R language [25,37], all of the radiomics features were fed into the LASSO regression model [15]. We applied a 10-fold cross-validation in the training set to find optimal λ to avoiding model simplification and overfitting.

#### 4.3.2. Identifying the Best Performing Model and Construction of Risk Score for Survival Prediction

From the radiomics features identified by LASSO in the training set, we developed and compared univariate Cox proportional hazard models to identify the radiomics signatures associated with overall survival time in the training set using the survival package of the R language (statistical significance was determined when the *p*-value was less than 0.05). With this prognostic radiomics signatures model, we generated risk score by the formula, previously described in [15,38,39,40,41]:

Risk Score = $\sum_{i=1}^{n}\left(\beta \times radiomic\;signature\;value\right)$, the number *n* stands for the number of radiomics signatures features, β stands for regression coefficient of each radiomics signature obtained from the multivariate Cox proportional hazard model (Table 3). We derived the risk score by a multivariate Cox proportional hazard model by using the Est.PH function in survC1 package of the R language. Based on this package, each patient’s risk score was determined.

To ensure consistency and independent predictability of radiomics signatures features, which aids in the maximization of their predictive potential, we conducted correlation and tumor volume dependence analysis, as some recommend and authors research [42,43,44,45]. A high correlation between radiomics signature features and tumor volume indicated that they were proxy for tumor volume and were not valuable for overall survival prediction. In contrast, if their correlation was low, they can signify independent predictability for survival prediction. Shafiq-ul-Hassan et al. [44] showed that correlation values of r > 0.9 defined high correlation. Fave et al., chose a threshold r > 0.95 [43]. In our study, we chose a threshold of 0.75, and we evaluated each pair of variables calculated by Pearson’s correlation analysis (Aerts et al. [42]) to identify the correlation between 10 radiomics signatures features, risk score, and tumor volume.

Finally, in the testing and validation sets, we locked and independently assessed the risk score created by the best radiomics signatures model for overall survival prediction.

### 4.4. Statistical Analysis

We used a Wilcoxon rank-sum test for continuous variables and a χ2 test for categorical variables to determine demographic differences between the training set and the testing set. All radiomics features were transformed by z-score transformation before building the model. With the best model radiomics signatures, we calculated a risk score for each patient. Then patients were separated into high- and low-risk groups based on the median of the risk score. We used the risk score generated from the multivariate Cox proportional hazard model to evaluate survival ability estimation. In the training set, the Kaplan–Meier curve dichotomized groups into high and low-risk groups by the median of the risk score. The time-independent receiver operating characteristic curve (ROC) was used to evaluate the risk prediction ability of the risk score model for overall survival (R package, survival-ROC function, version 1.0.3) compared to clinical parameters and evaluate the prediction ability of the risk score model in 1, 3, and 5-year survival in the training set. We locked and independently assessed the risk score that was built from the signatures model in the testing set and validation set. R (version 3.3.0) and Python (version 3.8) were used to do all of the analyses (statistical significance was defined as a *p*-value of less than 0.05).

## 5. Conclusions

In this study, the risk score developed from the CT-based radiomics signatures has great potential to predict overall survival in NSCLC patients. Furthermore, because the risk score model may differentiate patients into high-risk and low-risk groups, patients in the high-risk group have a worse overall survival rate than patients in the low-risk group. It has the potential to improve prognostic prediction accuracy as compared to clinical parameters. More research is required to confirm the models’ generalizability on external validation datasets.

## Figures and Tables

**Figure 1 cancers-13-03616-f001:**
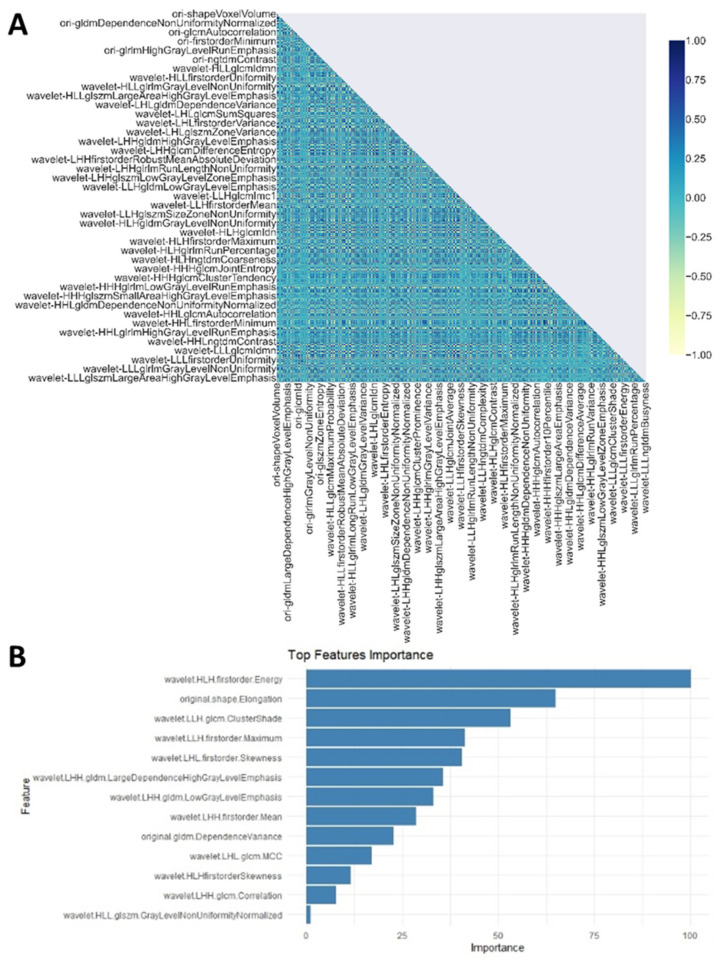
Features’ selection. (**A**) Removing highly correlated features in the training set. (**B**) Thirteen top features selected by LASSO in the training set.

**Figure 2 cancers-13-03616-f002:**
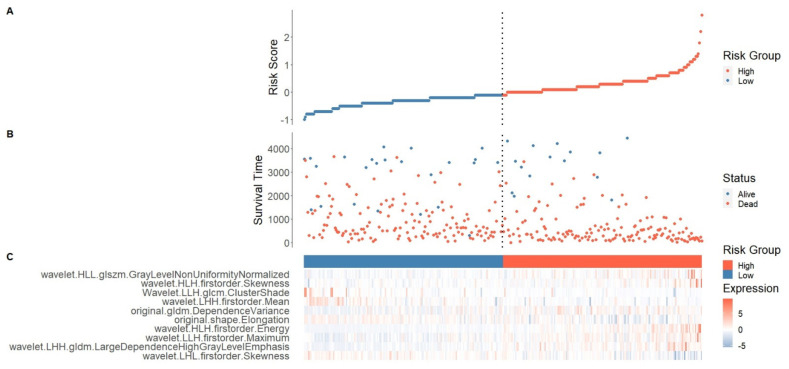
Risk score distribution and ten features radiomics signatures expression by Multivariate COX regression in the training set. (**A**) Risk score distribution. (**B**) Scatter plot divide patients into two groups: low-risk group (steel blue color) and high-risk group (tomato red color). (**C**) Expression heat map of 10 radiomics signatures in the training set.

**Figure 3 cancers-13-03616-f003:**
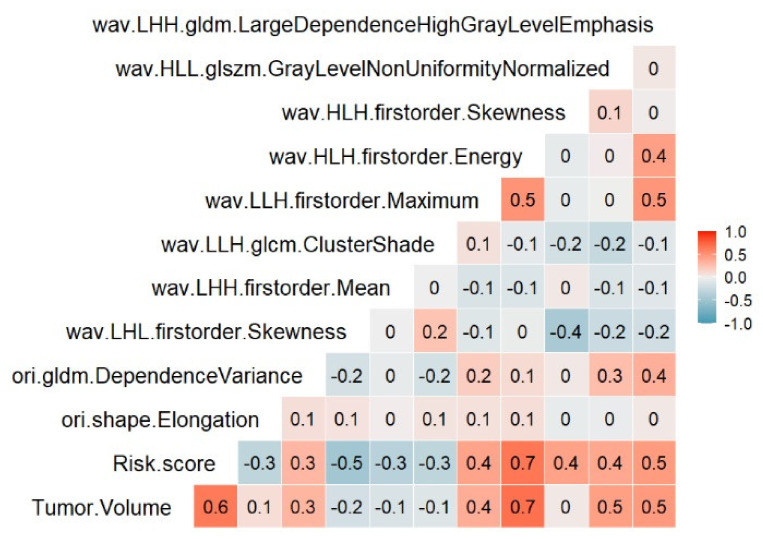
Correlation between ten radiomics signature features, Risk score, and tumor volume in the training set.

**Figure 4 cancers-13-03616-f004:**
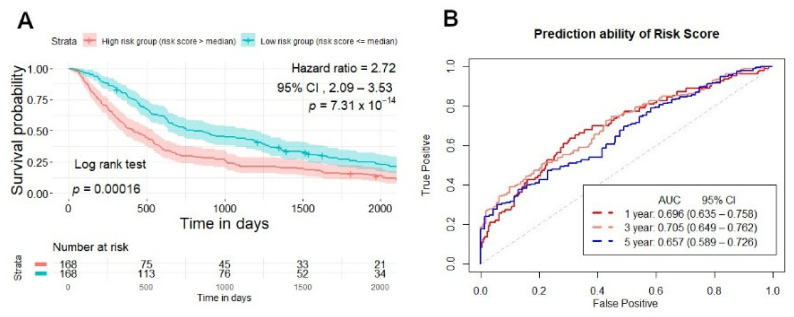
Survival prediction of risk score generated from CT-based radiomics signatures in training set. (**A**) Kaplan–Meier curves stratified patients into two groups: high and low-risk group by risk score in the training set. (**B**) ROC curves illustrated the prediction ability of risk score for overall survival prediction in 1, 3, 5 years in the training set.

**Figure 5 cancers-13-03616-f005:**
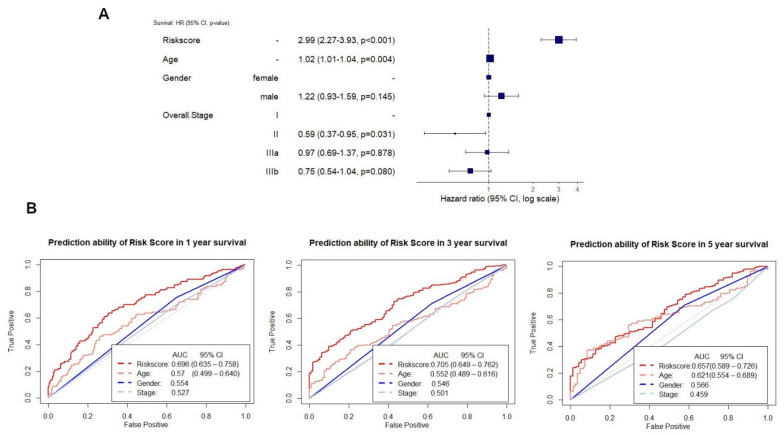
Risk score is an independent prognostic factor for survival prediction compared to clinical features in the training set. (**A**) Multivariate Cox regression showed that risk score was an independent prognostic factor for survival prediction compared to clinical features. (**B**) ROC curves compare risk score and clinical parameters (age, gender, stage) for survival prediction in 1 year, 3 years, and 5 years in the training set.

**Figure 6 cancers-13-03616-f006:**
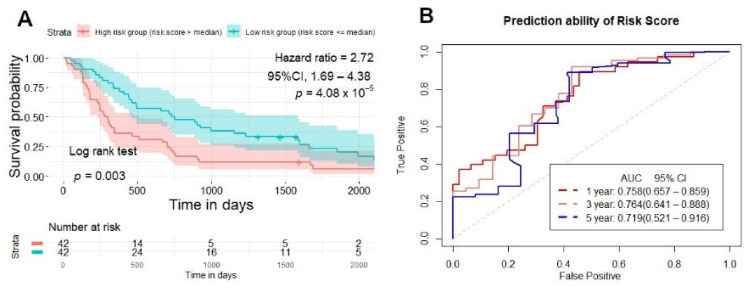
Survival prediction of risk score generated from CT-based radiomics signatures in the testing set. (**A**) Kaplan–Meier curves stratified patients into two groups: high and low-risk group by risk score in the testing set. (**B**) ROC curves illustrate the prediction ability of risk score for overall survival prediction in 1, 3, 5 years in the testing set.

**Figure 7 cancers-13-03616-f007:**
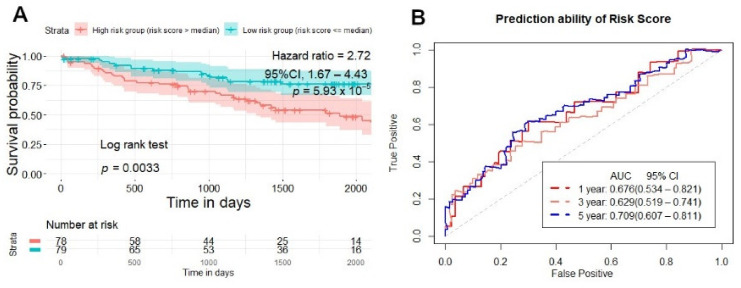
Survival prediction of risk score generated from CT-based radiomics signatures in the validation set. (**A**) Kaplan–Meier curves stratified patients into two groups: high and low-risk group by risk score in the validation set. (**B**) ROC curves illustrate the prediction ability of risk score for overall survival prediction in 1, 3, 5 years in the validation set.

**Figure 8 cancers-13-03616-f008:**
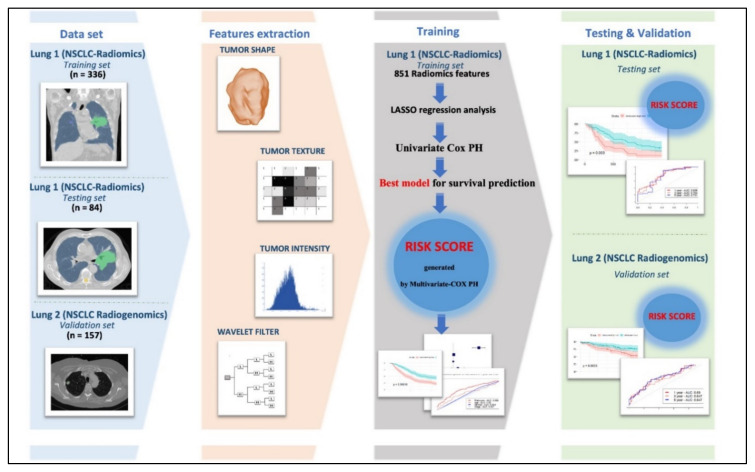
Flow chart of identified radiomics signatures and build the Risk score for survival prediction of the study.

**Table 1 cancers-13-03616-t001:** Baseline patient characteristics.

Lung 1 (NSCLC-Radiomics)				Lung 2 (NSCLC Radiogenomics)		
	Training Set	Testing Set			Validation Set	** *p*-Value
Parameter	(*n* = 336)	(*n* = 84)	* *p*-value	Parameter	(*n* = 157)	
Age (mean SD)	67.67 (10.04)	69.36 (10.19)	0.183	Age (mean SD)	68.97 (9.52)	0.302
Gender				Gender		
Men (%)	229 (68.2)	60 (71.4)	0.654	Men (%)	109 (69.4)	0.967
Type of NSCLC (%)			0.545	Type of NSCLC (%)		<0.001
Adenocarcinoma	40 (13.3)	11 (14.1)		Adenocarcinoma	126 (80.3)	
Large cell	95 (31.7)	19 (24.4)		Squamous cell	28 (17.8)	
Squamous cell	115 (38.3)	36 (46.2)		Nos	3 (1.9)	
Nos	50 (16.7)	12 (15.4)		Stage (%)		<0.001
Stage (%)			0.126	Tis	6 (3.8)	
I	77 (23.0)	15 (17.9)		I	86 (54.8)	
II	30 (9.0)	10 (11.9)		II	23 (14.6)	
IIIa	95 (28.4)	16 (19.0)		IIIa	15 (9.6)	
IIIb	133 (39.7)	43 (51.2)		IIIb	5 (3.2)	
IV	0	0		IV	4 (2.5)	
Survival time (median(IQR))(days)	579.00 (286.25, 1510.25)	456.50 (229.00, 1038.50)	0.073	Survival time (median (IQR))(days)	1315.00 (630.00, 1921.00)	<0.001

* *p*-value determined demographic differences between the training set and the testing set; ** *p*-value determined demographic differences between the Lung 1 (NSCLC-Radiomics) data set and the Lung 2 (NSCLC Radiogenomics) data set.

**Table 2 cancers-13-03616-t002:** Univariate Cox regression analysis to identifying the radiomics signatures in the training set.

Feature	Hazard Ratio	*p*-Value	Concordance
original.shape.Elongation	0.49 (0.25–0.95)	0.04 *	0.54 (se = 0.019)
original.gldm.DependenceVariance	1.01 (1.00–1.02)	0.02 *	0.55 (se = 0.017)
wavelet.LHL.glcm.MCC	2.71 (0.75–9.88)	0.13	0.517 (se = 0.018)
wavelet.LHL.firstorder.Skewness	0.70 (0.56–0.90)	<0.01 *	0.57 (se = 0.018)
wavelet.LHH.gldm.LargeDependenceHighGrayLevelEmphasis	1.00 (1.00–1.01)	<0.001 *	0.57 (se = 0.017)
wavelet.LHH.gldm.LowGrayLevelEmphasis	18.82 (0.00–1662852)	0.61	0.48 (se = 0.017)
wavelet.LHH.firstorder.Mean	0.85 (0.74–0.98)	0.02 *	0.54 (se = 0.017)
wavelet.LLH.glcm.ClusterShade	1.0 (0.99–1.00)	0.04 *	0.54 (se = 0.018)
wavelet.LLH.firstorder.Maximum	1.00 (1.00–1.01)	<0.01 *	0.57 (se = 0.018)
wavelet.HLH.firstorder.Energy	1.00 (1.00–1.01)	<0.001 *	0.58 (se = 0.018)
wavelet.HLH.firstorder.Skewness	1.48 (1.10–1.99)	<0.01 *	0.513 (se = 0.019)
wavelet.LHH.glcm.Correlation	0.02 (0.00–1.40)	0.07	0.54 (se = 0.017)
wavelet.HLL.glszm.GrayLevelNonUniformityNormalized	130.61 (3.30–5169)	<0.01 *	0.53 (se = 0.018)

* Statistically significant (*p* < 0.05).

**Table 3 cancers-13-03616-t003:** Multivariate Cox regression analysis of ten radiomics signatures in the training set.

Feature	β Coefficient	Hazard Ratio	*p*-Value
original.shape.Elongation	−9.936 × 10^−1^	0.37 (0.18–0.75)	0.006
original.gldm.DependenceVariance	5.230 × 10^−3^	1.01 (0.99–1.02)	0.349
wavelet.LHL.firstorder.Skewness	−1.693 × 10^−1^	0.84 (0.66–1.08)	0.184
wavelet.LHH.gldm.LargeDependenceHighGrayLevelEmphasis	4.764 × 10^−6^	1.00 (1.00–1.00)	0.627
wavelet.LHH.firstorder.Mean	−1.489 × 10^−1^	0.86 (0.73–1.01)	0.073
wavelet.LLH.glcm.ClusterShade	−6.460 × 10^−6^	1.00 (1.00–1.00)	0.654
wavelet.LLH.firstorder.Maximum	1.074 × 10^−4^	1.00 (1.00–1.00)	0.734
wavelet.HLH.firstorder.Energy	8.738 × 10^−9^	1.00 (1.00–1.00)	<0.001
wavelet.HLH.firstorder.Skewness	3.567 × 10^−1^	1.43 (1.06–1.93)	0.02
wavelet.HLL.glszm.GrayLevelNonUniformityNormalized	2.988	19.85 (0.36–1108.62)	0.145

## Data Availability

Public data can be freely accessed and downloaded at https://wiki.cancerimagingarchive.net/display/Public/NSCLC-Radiomics (accessed on 7 January 2021) and https://wiki.cancerimagingarchive.net/display/Public/NSCLC+Radiogenomics (accessed on 7 January 2021).

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
