# Peer review of "Risk Score Generated from CT-Based Radiomics Signatures for Overall Survival Prediction in Non-Small Cell Lung Cancer"

_cancers, 2021, doi:10.3390/cancers13143616_

Round 1
Reviewer 1 Report
Please see attached Word document.

Reviewer 2 Report
- 'Introduction' section. The first time you mention computed tomography you should report the acronym (CT) immediately after (line 60 page 2).
- There are some typos and language errors in the paper; please check all the paper again and correct (e.g. Fig legend 4 '..cox regression show..' and '..survival prediction compare to...'
- Please discuss the possibility to apply this proposed risk score to a low-dose screening CT for lung cancer early detection. There are some large trials on-going on this aim. Consider these ref: -Computed Tomography Screening for Early Lung Cancer, COPD and Cardiovascular Disease in Shanghai: Rationale and Design of a Population-based Comparative Study. Acad Radiol. 2021 Jan;28(1):36-45. doi: 10.1016/j.acra.2020.01.020. Epub 2020 Mar 6. PMID: 32151538. -Low-Dose Computed Tomography Screening Proposal for the "Big-3 Diseases": Lung Cancer, Chronic Obstructive Pulmonary Disease, and Cardiovascular Disease. Acad Radiol. 2021 Jan;28(1):46-48. doi: 10.1016/j.acra.2020.07.035. Epub 2020 Aug 15. PMID: 32807607.
- The application of radiomics to that screening should offer adjunctive advantages.
Round 2
Reviewer 1 Report
All issues and comments have been resolved.
There are now further requests.
Author Response
Thanks for your positive comments.
Reviewer 2 Report
I am satisfied with the revisions performed.
Author Response
Thanks for your positive comments.